# Significance of Type II Collagen Posttranslational Modifications: From Autoantigenesis to Improved Diagnosis and Treatment of Rheumatoid Arthritis

**DOI:** 10.3390/ijms24129884

**Published:** 2023-06-08

**Authors:** Tsvetelina Batsalova, Balik Dzhambazov

**Affiliations:** Faculty of Biology, Paisii Hilendarski University of Plovdiv, 24 Tsar Assen Str., 4000 Plovdiv, Bulgaria; tsvetelina@uni-plovdiv.bg

**Keywords:** autoimmunity, collagen type II, posttranslational modifications, rheumatoid arthritis

## Abstract

Collagen type II (COL2), the main structural protein of hyaline cartilage, is considerably affected by autoimmune responses associated with the pathogenesis of rheumatoid arthritis (RA). Posttranslational modifications (PTMs) play a significant role in the formation of the COL2 molecule and supramolecular fibril organization, and thus, support COL2 function, which is crucial for normal cartilage structure and physiology. Conversely, the specific PTMs of the protein (carbamylation, glycosylation, citrullination, oxidative modifications and others) have been implicated in RA autoimmunity. The discovery of the anti-citrullinated protein response in RA, which includes anti-citrullinated COL2 reactivity, has led to the development of improved diagnostic assays and classification criteria for the disease. The induction of immunological tolerance using modified COL2 peptides has been highlighted as a potentially effective strategy for RA therapy. Therefore, the aim of this review is to summarize the recent knowledge on COL2 posttranslational modifications with relevance to RA pathophysiology, diagnosis and treatment. The significance of COL2 PTMs as a source of neo-antigens that activate immunity leading to or sustaining RA autoimmunity is discussed.

## 1. Introduction

Rheumatoid arthritis (RA) is a systemic autoimmune disorder with complex etiology comprising various genetic and environmental factors, as well as age-related alterations [1,2]. The disease affects different organs and systems in the organism (cardiovascular system, lungs, kidneys, nervous system, exocrine glands, gastrointestinal system, etc.) [3] and increases the risk of certain malignancies, but the major target of the autoreactive pathological process is the synovia of peripheral joints [4]. RA causes chronic synovitis and hyperplasia due to infiltration of inflammatory cells in joints, activation of resident cells and production of proinflammatory cytokines and matrix-destructive enzymes [5,6]. If an adequate and timely treatment is not applied, these processes lead to destruction of normal cartilage structure followed by pannus formation and bone erosion associated with marked physical disability and morbidity [7,8].

A significant proportion of the population worldwide suffers from RA (between 0.3 and 1%), which determines it as one of the most common autoimmune diseases [9,10]. Recent therapeutic strategies applying conventional synthetic disease-modifying antirheumatic drugs (synthetic DMARDs: methotrexate, sulfasalazine, leflunomide, etc.), biological agents (biological DMARDs: tumor necrosis factor α inhibitors, biologics that target interleukins (IL-1, IL-6), T-cells, B-cells or growth and differentiation factors (receptor activator of nuclear factor kappa-Β ligand, RANKL; granulocyte-macrophage colony-stimulating factor, GM-CSF)), targeted synthetical agents (Janus-activated kinase inhibitors) and combination therapies have significantly improved the symptomatic treatment of RA and led to a prolonged amelioration of disease signs [2,11]. However, certain RA patients do not respond to these therapies [6]. Another critical issue is the various adverse effects caused by DMARDs and new treatment strategies [12]. Specifically, the application of biological agents is associated with immune suppression and increased risk of infections and malignancies. These facts prove the need for the development of precision medicine therapies based on the identification of sustainable disease biomarkers. However, considering the complex nature and etiology of the RA establishment of immune-specific treatment approaches is still an unsolved task [11,13]. Numerous studies prove that the autoimmune response causing RA can be directed against different native autoantigens [14] and posttranslationally modified self-proteins [15,16]. Some of them are ubiquitously expressed—glucose-6-phosphate isomerase (G6PI) [17], heterogeneous nuclear ribonucleoprotein-A2 (RA33) [18,19], immunoglobulin binding protein (BiP) [20], and citrullinated proteins (fibrinogen, filaggrin, vimentin, α-enolase, tenascin-C, histones, etc.) [21,22,23]—while others are joint-specific—COL2 [24,25,26], aggrecan [27,28], human cartilage glycoprotein 39 [29], and cartilage oligomeric matrix protein (COMP) [30]. Considering the heterogeneous nature of RA, it is possible that different antigens cause and perpetuate the disease in patients from different subgroups. However, a significant number of scientific publications suggest that COL2 holds an important role in RA development and pathogenesis. The evidence supporting this hypothesis is: the immunization of experimental animals with COL2 induces disease (collagen-induced arthritis, CIA) [31] similar in pathology to RA [32]; the presence of anti-COL2 autoantibodies in serum and synovial fluid in a significant proportion of RA patients [33,34]; anti-COL2 autoantibodies have been suggested to play an important role in both the early phase of RA [25] as well as in disease progression [25,35]; COL2-reactive T lymphocytes are present in the peripheral blood and synovial fluid of RA patients [36,37,38,39,40]; the COL2-specific T-cell response in RA patients correlates with inflammasome activity and the extent of radiographic changes in the joints [40]; CIA development can be inhibited by treatment with COL2 or collagen peptides [41,42,43,44]; COL2 therapy alleviates the symptoms and progression of RA [45,46,47].

COL2 belongs to the group of fibril-forming collagens [48]. Its molecule is a homotrimer of three identical α1 (II) polypeptide chains containing glycine in every third position of its amino acid sequence and assembling in a left-handed triple helix. Following production and exocytosis in the extracellular matrix (ECM), COL2 molecules assemble in a regular staggered array to make a fibril with characteristic cross-striations [49]. COL2 is expressed in a tissue-specific manner and it is the main protein of the hyaline cartilage, vitreous humor of the eye and nucleus pulposus [50]. In addition to its structural functions, COL2 is involved in extracellular matrix remodeling, as well as interactions with different cell types and control of their activities [51].

During biosynthesis, after translation by ribosomes on the rough endoplasmic reticulum, collagen polypeptide chains undergo extensive modifications, which are crucial for proper folding and homotrimeric molecule formation and its functionality [52]. Proline and lysine residues are hydroxylated and glycosylated before joining the polypeptide chains in procollagen triple helix, and subsequently, disulfide bonds are formed [48]. This is followed by the secretion of procollagen chains in the extracellular matrix where the amino- and carboxy-terminal globular regions of the molecules are removed by specific proteases [53]. Then, COL2 triple helices self-assemble to form collagen microfibrils that are stabilized by the oxidation of lysine side chains, leading to hydroxylysyl pyridinoline and lysyl pyridinoline cross-links [54]. COL2 cross-links stabilize the large collagen fibrils, assuring their proper mechanical properties [53]. Consequently, posttranslational modifications (PTMs) play a major role in COL2 formation and fibril assembly, which is crucial for normal joint structure and stability. Apart from their important physiological role, various COL2 PTMs can be generated during autoimmunity or under the influence of specific environmental factors, including infectious agents that lead to RA. Therefore, the purpose of this review article is to summarize the knowledge on different COL2 PTMs implicated in RA pathogenesis. 

## 2. Collagen Type II (COL2) Posttranslational Modifications in Rheumatoid Arthritis

By definition, PTMs represent a large group of covalent chemical reactions that alter the primary structure of the proteins [55]. They are carried out after the synthesis of polypeptide chains or during translation. To date, more than 400 different PTMs are identified [56] and they provide a major mechanism for the diversification of cellular proteome, which impacts a variety of physiological processes by modulating protein structure, stability, activity, localization and specific properties [57]. Normal cellular events associated with aging and pathological conditions, such as infections, trauma and others, increase the extent of posttranslational protein alterations [58]. Furthermore, certain PTMs or their dysregulation could negatively affect normal cellular functions and lead to pathological processes. In the context of autoimmunity, PTMs are responsible for generation of altered self-proteins or neo-epitopes that induce disease [59,60]. This biological phenomenon has been well described for RA pathogenesis [61,62]. A typical example is citrullination. The detection of autoantibody reactivity against citrullinated proteins [63] has improved disease diagnosis and it has been included in the standard classification criteria for RA for more than a decade [64]. In addition to anti-citrullinated protein antibodies (ACPA), autoantibody relativities towards other PTMs (i.e., carbamylation, acetylation, glycosylation, oxidative modifications, etc.) have been demonstrated [61,65]. Thus, it is now generally accepted that anti-modified protein antibodies (AMPA) play a significant role for both RA diagnosis and pathology. Steen et al. have demonstrated that AMPA from RA patients are cross-reactive and recognize specific posttranslationally modified amino acid motifs instead of certain protein, which suggests that AMPA reactivity appears at different sites and different time points during RA development, eventually targeting joints and leading to destruction and disability [66]. In addition, T-cell responses directed towards posttranslationally modified proteins have been documented [37,62,67]. It has been proven that neo-epitopes generated by PTMs escape central tolerance mechanisms [68] and could induce autoimmune disease. Anti-PTMs reactivities during RA and a prior manifestation of clinical signs of disease include responses against the main structural protein of joints—COL2. The first scientific reports implying that specific PTMs play a role in autoimmune arthritis were published more than two decades ago [22,69,70]. Since then, a significant amount of experimental data on the topic have been gathered, and a number of different PTMs of the COL2 molecule have been discovered and associated with RA development [61,62,69,71,72] (Figure 1). The main types of COL2 PTMs are discussed in the following subsections.

### 2.1. Hydroxylation

The complex multi-step process of COL2 biosynthesis involves the hydroxylation of specific amino acid residues, which plays an important role in the formation and stabilization of the triple helical molecular structure, as well as the stabilization of COL2 fibrils [54]. Prolyl and lysyl residues bound to N-terminally directed glycine residues can be hydroxylated by prolylhydroxylases and lysylhydroxylases, and hence, converted to hydroxyproline (HyP) and hydroxylysine (HyK) [73,74]. HyP residues contribute to the thermal stability of collagen molecules and their extent does not change significantly under physiological conditions. However, the numbers of Hyl residues vary considerably under both the normal and pathological states [54]. An important physiological role of COL2 hydroxylation was demonstrated by Tenni et al. who showed that lysine and HyK residues are essential for decorin binding to collagen [75]. These results are relevant to the potential implication of COL2 hydroxylation in joint-directed disease because decorin-collagen interactions are involved in the integrin-independent control of cellular activity and extracellular matrix (ECM) stability [75,76].

In addition, Notbohm et al. have shown that the degree of lysine hydroxylation and HyK glycosylation may influence collagen fibril formation and morphology [77]. Thus, it could be suggested that altered COL2 hydroxylation (lysyl hydroxylation) may affect cartilage structure, which is of central importance for supporting normal joint function. Concordantly, it has been determined that the strength of the COL2-specific immune response in CIA and its arthritogenicity depends on and correlates with the extent of lysine hydroxylation, as well as hydroxylysine glycosylation [78].

PTMs, particularly hydroxylation, can be present to a different extent also in the T-cell immunodominant epitope from COL2 identified in studies of collagen-induced arthritis and RA [38,69]. The hydroxylation of lysine residue in position 264 in this epitope has been determined as a transient PTM form that could contribute to the development of COL2 autoreactivity, as Yamada et al. have shown that the glycosylated variant of the respective epitope preponderates in human and rat cartilage COL2 [79]. Therefore, the hydroxylated form could escape tolerance induction and increase the risk of joint-specific autoimmunity. This hypothesis is supported by further research that demonstrated a PTMs pattern of the COL2_259–273_ epitope [72] that was altered by arthritis. However, the major differences were associated with the reduced glycosylation pattern of K264, and thus, the role of lysyl hydroxylation in this context needs to be further investigated. 

### 2.2. Glycosylation 

The degree of hydroxylation and glycosylation of COL2 varies because these processes are limited in time and take place during the period of synthesis of collagen polypeptide chains to form the triple helical collagen molecule. It also depends on the functional state of chondrocytes in cartilage tissue [73]. Hydroxylysine residues can be galactosylated (Gal-HyK). Subsequently, α-glucopyranose can bind to Gal-HyK residue, forming α-glucopyranosyl-(1→2)-β-galactopyranosyl-hydroxylysine (Glc-Gal-HyK). More than two decades ago, Michaelsson et al. have shown that COL2 glycosylation is a crucial PTM type that influences COL2 arthritogenicity in mice [80]. These pioneering experiments demonstrated that immunization with carbochidrate-depleted COL2 leads to reduced arthritis severity, incidence and time of onset compared to CIA caused by glycosylated collagen [80]. They also indicated a major role for T cell reactivity towards glycosylated COL2 because immunization with deglycosylated protein did not affect COL2-specific antibody responses. It has been suggested that O-linked carbohydrates, particularly those limited to one and two monosaccharide residues, may be a prerequisite for T-cell recognition of COL2 in autoimmune arthritis [69]. More complex carbohydrate moieties could avert the effective interaction of the trimolecular complex: major histocompatibility complex class II (MHC II)—autoantigenic COL2 peptide—T-cell receptor (TCR). An immunodominant COL2 epitope comprising amino acid residues from positions 256 to 270 was initially identified based on experiments with mouse models [81] and glycosylation of lysine in position 264 was subsequently pointed to as a crucial determinant of COL2-specific T-cell reactivity implicated in CIA development [69,80]. These findings were followed by the definition of a determinant core (COL2_263–270_) needed for interaction with MHC II molecules bearing the shared peptide binding pocket (HLA-DRB1 amino acid positions 70–74) associated with increased risk of RA development [82]. This preferential binding to HLA-DR4 and DR1 molecules supported the potential role of COL2 reactivity in autoimmune arthritis [83,84]. The amino acid sequence of the COL2 immunodominant epitope, including positions 259/260 to 273, was defined by fine investigations of contact points with MHC and TCR [85]. Concordantly, collagen-specific T lymphocyte responses directed towards the same immunodominant epitope were identified in RA patients [36,37,38,86,87]. Moreover, T-cell reactivity against the glycosylated in position K264 COL2 (Gal264) epitope was detected during RA development [37,38,67]. Although the immunodominant epitope contains two lysine residues (K264 and K270) glycosylation in position 270 has been shown to elicit weaker T-cell responses compared to the reactivity against the Gal264 epitope both in RA patients and mouse models [37,88]. Recent crystallographic studies and molecular modelling revealed the posttranslationally galactosylated lysine 264 as a major contact point for the TCR, and hence confirmed its potential role for arthritogenic and tolerogenic COL2-specific responses [67,89]. 

Glycosylation is one of the most common posttranslational modifications in mammalian cells [90], and collagens are representative examples of proteins bearing this PTM to a high degree [53]. Moreover, collagen structure and function are strongly dependent on the proper pattern of hydroxylation and subsequent glycosylation of significant amount of lysine residues [52]. Therefore, it is not surprising that the immunodominant COL2 epitope in normal human joint cartilage tissue has been found to be glycosylated at K264, while both glycosylated and non-glycosylated forms were detected in arthritic cartilage [72]. This indicates that an altered or variable PTM pattern during the disease course could be involved in arthritis development depending on the structures presented to the immune system. In accordance to this finding, COL2-specific T-cells from RA patients react against both the nonmodified and glycosylated in position 264 epitope and display an inflammatory response [37]. Longitudinal evaluations demonstrated that these reactivities persist during the disease course and show variable magnitude and fine specificity, including other modified forms of the epitope. Therefore, it is clear that reactivity against the COL2 immunodominant epitope and its glycosylated form could contribute to RA development and/or perpetuate the disease. Hence, immunity against glycosylated COL2 in different subgroups of RA patients could form the basis for COL2 PTM-specific therapeutic approaches.

### 2.3. Citrullination

The enzymatic conversion of arginine to citrulline mediated by peptidyl arginine deiminases (PADs) plays an important role in normal physiological processes, including apoptosis, terminal epidermal differentiation, embryonic development, regulation of gene expression, hair growth, immune responses and others [91,92]. However, the substitution of a positively charged alkaline amino acid with a neutral one affects the charge, isoelectric point, ionic and hydrogen bonding properties of the protein, which could influence its function, interaction with other proteins and half-life [93]. Therefore, citrullination has been implicated in the pathogenesis of different diseases, including many autoimmune diseases, such as multiple sclerosis, type I diabetes, systemic lupus erythematosus, ulcerative colitis and rheumatoid arthritis [92,94]. In fact, citrullination is one of the most important PTMs in RA regarding both disease pathogenesis and diagnostics [22,63]. ACPA represent a serological marker for RA with more than 90% specificity. They precede the onset of clinical symptoms [95] and are predictive of the development of more destructive disease phenotypes [96,97]. ACPA reactivity against both ubiquitous, tissue- and inflammation-restricted citrullinated proteins have been detected in autoimmune arthritis [95]. Recently, more than 150 citrullinated cellular and extracellular proteins have been identified in RA synovia, comprising the so-called RA-associated citrullinome, which has been suggested to play an essential role in disease pathogenesis and the generation of ACPA [94,98]. Despite this variety of detected citrullinated antigens, autoimmunity towards COL2 that has been modified by citrullination has been well determined in the RA disease course [99,100,101,102,103]. Anti-citrullinated COL2 antibodies can induce and exacerbate arthritis in experimental murine models [100,104,105]. Citrullinated COL2 has been detected in the synovial fluid and cartilage of RA patients [105,106] and has been shown to induce autoimmune arthritis in experimental animals [107,108]. Humoral immunity against different citrullinated collagen epitopes has been detected in RA patients with more than 90% specificity [109,110]. Snir et al. reported that antibodies recognizing citrullinated COL2 C1 epitope were elevated in RA synovial fluid [111]. Subsequently, Brink et al. showed that the frequency of anti-cit C1 COL2 antibodies is similar to anti-carbamylated protein antibodies in a cohort of more than 250 individuals [112]. Antibody reactivity against cyclic citrullinated COL2 alpha chain epitopes has also been detected in RA patients [102]. Autoantibodies from RA patients’ sera recognizing citrullinated COL2 have been shown to bind to arthritic CIA and RA joint cartilage [101,106]. These findings support that anti-citrullinated COL2 humoral reactivity plays a role in joint inflammation and cartilage destruction. Furthermore, circulating citrullinated COL2 has been detected in RA patients’ sera and its levels correlated positively with RF and ACPA [104]. Intriguingly, a number of RF- and ACPA-negative RA sera also show anti-citrullinated COL2 reactivity, which highlights an opportunity for optimization of current diagnostic assays for RA, including ACPA-negative patients [104]. 

Anti-COL2 ACPA responses have been associated with the HLA-DRB1*04 shared epitope (SE) alleles [113]. The citrullination of COL2_359–369_ epitope has been shown to be sufficient to activate B-cell immunity in HLA-DRB1*04 subjects [114]. However, a recent report by Too et al. showed that populations from different geographic regions differ in ACPA fine specificities, and in a Malaysian population, the relationship between COL2 responses and HLA-DRB1 SE alleles was weaker compared to a group of Swedish RA patients included in epidemiological investigation of an RA (EIRA) study [99]. Interestingly, antibodies to citrullinated COL2 peptide were significantly elevated in the Malaysian patients, suggesting that genetic and environmental factors could influence ACPA fine specificities. ACPA reactivities are complex and include COL2-specific antibodies with a narrow recognition pattern, as well as highly cross-reactive immunoglobulins that bind to different citrullinated epitopes sharing a certain structural motif [66,100,102,115]. 

COL2 citrullination has been shown to reduce cellular adhesion mediated by α10β1 and α11β1 integrin dimers, which affected the survival of mesenchymal stem cells [116]. On the other hand, the COL2 binding capacity of integrins expressed by inflammatory cells was not influenced or minimally affected. These results lead to the hypothesis that the citrullination of joint collagen determines a specific pattern of interaction with different cell types, potentially favoring the binding of cells with proinflammatory properties and preventing cartilage localization and the survival of mesenchymal stem cells with immunosuppressive and anti-inflammatory potential. Another role of COL2 citrullination in RA that involves interactions with the leukocyte-associated immunoglobulin such as receptor 1 (LAIR-1; CD305) has recently been demonstrated [117]. Collagens are natural ligands for LAIR-1, which, when activated, have the potential to inhibit inflammatory responses and contribute to the maintenance of immune homeostasis [118]. Interestingly, Myers et al. have demonstrated that the PAD4-mediated citrullination of COL2 induces the antagonistic effect of the protein regarding its interaction with LAIR-1 [117]. Supported by the proven expression of PAD2 and PAD4 in RA synovia [119], these findings suggest that in arthritic joints, citrullinated COL2 could inhibit the LAIR-1-mediated downregulation of autoimmune responses, and thus, perpetuate the disease.

The effect of COL2 citrullination has been investigated in relation to MHC II presentation and T-cell reactivity. The conversion of arginine to polar-neutral citrulline affects antigenic epitope affinity to MHC II molecules and their recognition by T-cells that could even result in novel TCR interactions [61,120,121]. It has been suggested that polar amino acids bind the positively charged P4 SE pocket of HLA-DRB1*0401 and induce the activation of autoreactive T-cells [122]. However, using a panel of COL2 peptides, Sidney et al. showed that citrullination had relatively slight effect on binding to RA-associated HLA-DRB1*0401 and HLA-DRB1*0101 [121]. The authors suggested that the PTMs could rather affect T-cell recognition and possibly create new HLA-restricted T-cell epitopes. Indeed, such neo-epitope from citrullinated COL2 flanking amino acid residues 311–325, which was restricted to HLA-DRB1*1001, has been reported [123]. Recently, Becart et al. showed that citrullinated COL2 peptides’ binding affinity to RA-associated HLA molecules could not predict the induction of autoimmune T cell responses [62] and resulted in a variable pattern of T-cell proliferation that did not correlate with proinflammatory cytokine production. Therefore, the role of COL2 citrullination for autoimmune T-cell responses in RA remains to be elucidated. It is clear that COL2 citrullination during joint inflammation generates new antigenic epitopes and some of them could perpetuate and expand pathogenic immune responses.

ACPA responses in RA patients are heterogenic [124,125,126]. Sahlström et al. have shown that monoclonal ACPAs exert high cross-reactivity and recognize citrullinated epitopes in various proteins, including COL2, which represent consensus peptide motifs such as Gly-Cit, Cit-Gly, Arg-Cit-Asp [127]. A year prior to this report, structural studies confirmed that the Cit-Gly motif is essential for the recognition of citrullinated COL2 epitopes by a monoclonal antibody derived from RA patient [115], and amino acid residues flanking the citrulline site are important for specific antigen–antibody interactions. The cross-reactivity of the antibody was attributed to polar interactions mediated by residues at more distal positions in the peptide backbone [115]. In addition, COL2 citrullination could lead to epitope flexibility and a partially denatured state of the triple helical structure [100]. Native non-PTM epitopes that mimic such structure could be cross-recognized by anti-citCOL2 antibodies, providing another possible mechanism explaining ACPA polyreactivity [100].

ACPA could be primed by citrullinated proteins that are not located in the joints. Most likely, immune responses initiated in mucosal tissues due to periodontal infections, altered gut microbiota balance, lung inflammation, etc., inducing ACPA production [128]. Citrulline-specific reactivity could appear at different sites and different time points during RA development. In addition, it has been demonstrated that ACPAs in the peripheral blood and synovial fluid of RA patients support neutrophil extracellular trap (NET) formation [129] and release of PAD enzymes [130], which increases the milieu of citrullinated antigens that could drive the generation of new high-affinity ACPAs and/or amplify ACPA production in the synovium [131], thus forming a vicious cycle in RA development. It has been shown that the affinity maturation of ACPA-producing B-cells achieved by somatic hypermutations contributes to the epitope spreading and polyreactivity of anti-citrullinated protein responses [132]. Furthermore, B-cells and ACPA can display broad reactivity, expanding beyond citrullination and including other PTMs [133]. How these cross-reactive ACPA/AMPA evolve to specifically attack joint cartilage and COL2 remains to be deciphered. Undoubtedly, the fine determination of anti-citrullinated COL2 reactivity could contribute to RA subgroup classification, diagnosis and therapy.

### 2.4. Carbamylation

The production of autoantibodies directed towards posttranslationally modified proteins is a central characteristic of RA [103,134]. During the last decade, in addition to ACPA, anti-carbamylated protein (anti-CarP) antibodies were highlighted as important biomarker for RA [65,134,135,136], which can be detected in patients’ serum years before disease onset [137,138] and increase gradually just before and during the manifestation of clinical symptoms [95,112]. The predictive value of anti-CarP antibodies includes ACPA-negative patients and contributes to the definition of RA subgroups with increased joint destruction over the disease course independent of ACPA [112]. 

Carbamylation (or homocitrullination) is a non-enzymatic chemical modification reaction between cyanate in the form of isocyanic acid and ε-amino group of a lysine residue. It leads to the formation of a non-standard amino acid with an ureido group-containing side chain (-NH-CO-NH_2_) similar to citrulline but one methylene longer [139]. Carbamylation can be stimulated in vivo by myeloperoxidase (MPO), which links this type of PTM with inflammatory conditions and inflammatory-driven carbamylation of cartilage proteins [140]. In fact, MPO-mediated carbamylation has been suggested as a natural defense mechanism that is activated through inflammatory conditions [141]. During RA, MPO released by neutrophils in joints, converts thiocyanate into cyanate, and thus, could lead to the generation of carbamylated neo-antigens, including carbamylated epitopes from COL2. It has been shown that homocitrulline-containing proteins are present in RA foot joints together with citrullinated proteins [142]. On the other hand, uremia and exposure to cigarette smoke enhance cyanate levels in the organism that could induce substantial carbamylation [143]. Therefore, similar to citrullination, carbamylation could take place in different sites in the organism, inducing anti-CarP immunity that eventually attacks joint cartilage. This hypothesis is supported by the fact that anti-CarP antibodies develop years before disease onset and increase during manifestation of clinical signs of autoimmune arthritis in both mice and RA patients [112,144]. In addition, the immunization of experimental animals with carbamylated proteins has been shown to prime reactivity against both carbamylated and acetylated proteins [145] or against citrullinated proteins [146], suggesting that certain B-cell responses could diversify into various distinct AMPA responses. These findings are supported by investigations on monoclonal ACPA responses that showed epitope spreading following affinity maturation (discussed in Section 2.3) [132]. Anti-CarP antibodies detected in RA patients also show cross-reactivity [147,148], as has been demonstrated for different AMPA responses [103,133]. Confirming these findings, AMPA reactivities were recently shown to be highly dynamic and, in addition to cross-reactivity studies with mouse models, showed that their specificity can evolve over time following encounter with different PTMs [149].

Antibodies recognizing both citrullinated and carbamylated COL2 telopeptides were detected in the sera of RA patients [110]. Although these epitopes do not occur naturally, it is known that collagen can be carbamylated in vivo [114,140]. Homocitrulline residues gradually accumulate in tissues containing matrix proteins with long half-life, such as collagens [150], which impairs the mechanical properties of the tissue [151,152]. For instance, carbamylated collagen type I (COL1) shows defective fibrillogenesis [153], which could lead to alterations of tissue structure due to altered sensitivity to proteolytic enzymes [154,155], affect cell–matrix interactions [156] and disrupt tissue homeostasis, possibly inducing an autoimmune response. The effect of the carbamylation of COL2 T-cell immunodominant epitope has been investigated in monozygotic twins, discordant in RA and non-SE RA patients. It was shown that the modification did not alter COL2-specific immunity in a DR4+ RA patient, but was sufficient to induce T-cell activation and production of proinflammatory cytokines in a non-DR4+ patient with active disease [114]. These findings suggest that COL2 carbamylation could contribute to RA development and play a role in disease pathogenesis in particular patient subgroups. Further investigations on carbamylated COL2-directed immunity are needed to define the role of this PTM for RA pathogenesis and perpetuation. 

### 2.5. Oxidative Modifications

Multiple factors are involved in joint damage during RA—production of autoantibodies, release of proinflammatory cytokines and matrix metalloproteinases, and generation of free radicals. Reactive oxygen, nitrogen and chlorine species produced in inflamed joints represent an active trigger of oxidative PTMs in collagen and other proteins that could induce and/or enhance autoimmunity [157]. It has been demonstrated that different oxidative modifications increase the antigenicity of COL2 and its aggregation and/or proteolytic cleavage [158,159]. Specifically, COL2 modified by hydroxyl radical showed significantly higher immunogenicity and arthritogenicity in rat CIA model compared to native COL2 [160]. Antibody reactivity against chemically modified COL2 bearing different oxidative modifications (PTMs induced by reactive radicals present in the inflamed joint and glycation PTMs) was detected in serum samples from RA patients [159]. In addition, RA sera showed binding to synovial fluid proteins, corresponding to proteolytically degraded and/or aggregated COL2 [159]. Taken together, these findings lead to the suggestion that COL2 in inflamed joints could serve as a potent source of posttranslationally modified neo-epitopes stimulating RA autoimmunity. Antibodies directed to a particular type of oxidative modification of COL2—chlorination—have been highlighted as specific biomarkers for RA, especially prevalent in early RA patients and DMARDs nonresponders [161]. However, a recent study performed in a cohort of 325 RA patients, including patients with early diagnosed disease and patients with established RA, did not confirm these findings. Antibody responses against COL2 modified by hypochlorous acid were evaluated in this study, together with antibodies against two other collagen oxidative modifications that can be present in inflamed RA joint—glycation and nitration. The obtained results led to the conclusion that antibodies targeting these PTMs are absent or inconsequential in RA patients [162]. 

#### Glycation

One of the most studied PTMs, a type of oxidative modification, is glycation—nonenzymatic reaction of binding of sugars or their by-products to amino groups of proteins, leading to molecular rearrangements and the formation of advanced glycation end-products [163]. Like carbamylation, glycation modifications were shown to accumulate in matrix proteins with a long half-life, such as collagens, which leads to the progressive impairment of mechanical tissue properties [164] due to altered molecular interactions at the collagen fibril surface [165] and perturbed cell-extracellular matrix interactions [156]. Such alterations could eventually contribute to RA pathogenesis. Both PTMs, carbamylation and glycation, are associated with molecular aging and different pathologic conditions. A competitive accumulation with predominance for carbamylation as a prompter one-step modification was demonstrated in experiments focused on COL1 [163]. These findings could be related to joint COL2 supported by the well-known anti-CarB immunity, while evidence indicating the existence of anti-glycated collagen in RA development are scarce, despite the fact that glycation was proven in RA despite the absence of hyperglycemia [166] and increased levels of advanced glycation end products were detected in the serum, synovial fluid and urine of RA patients [167,168]. A recent report by Rodriguez-Martinez et al. showed evidence for the presence of antibodies specific to glycated COL2 in the sera of RA patients [162], but the magnitude of the detected signal was relatively small to conclude that immunity against this type of COL2 modification plays a significant role, suggesting a more restricted posttranslationally modified autoantigens repertoire in RA.

### 2.6. Deamidation

Glutamine residues from QXP sequences in the polypeptide chains of cellular and ECM proteins can be preferentially converted to glutamic acid by Ca^2+^-dependent enzymatic reaction catalyzed by tissue transglutaminase. The same enzyme can catalyze protein cross-linking by the formation of an isopeptide bind between glutamine and lysine residues. Interestingly, the T-cell immunodominant epitope from COL2 contains glutamine in position 267, which is part of sequence QGP, suggesting that it can by modified by tissue transglutaminase. Tissue transglutaminase has been implicated in inflammatory conditions [169], and thus, could contribute to COL2 modification during autoimmunity and RA pathogenesis. This hypothesis was confirmed by Dzhambazov et al., who showed that tissue transglutaminase can catalyze both the conversion of glutamine and cross-link reactions on the T-cell COL2 epitope [170]. In vivo administration of tissue transglutaminase exacerbates CIA and joint destruction in mice, suggesting that PTMs catalyzed by this enzyme could be involved in RA development, but the precise mechanism remains to be elucidated.

### 2.7. Candidate Modifications/Other Modifications

Recently, autoantibodies against acetylated proteins have become prominent members of the AMPA group implicated in RA development, improved disease classification and diagnosis [61,103,145,171]. Acetylation is a reversible enzymatic reaction of the binding of acetyl groups to free amines of lysine residues. The involvement of acetylated COL2 in joint-specific autoimmunity has not been demonstrated yet. 

Homocysteinylation is another PTM type that could be involved in RA pathogenic reactions and inflammatory processes [172]. COL2 modification by L-homocysteine-thiolactone has not been confirmed as a significant collagen PTM in assays with RA patients’ serum samples [162]. 

COL2 conformational changes can be induced by decreased pH detected during synovial inflammation [173,174]. It was demonstrated that COL2 is more immunogenic in acidic environment and its pH-induced conformational changes correlate with arthritogenic properties [175]. These data represent another line in COL2 PTMs that can be involved in RA pathogenesis and remains to be further studied.

In summary, the dynamics of known COL2 PTMs associated with RA is presented in Figure 2. 

## 3. COL2 PTMs Contribution to RA Diagnosis and Treatment

The beneficial effect of COL2 therapeutic administration were shown already in the early 1990s [46,176,177]. To date, there are a number of studies on antigen-specific tolerance induction using COL2 or COL2 peptides in both experimental models and RA patients showing variable results and moderate-to-mild effects [45,47,178,179,180]. Increasing evidence for immunity directed towards different PTMs during RA development provided the basis for research on the efficacy of posttranslationally modified COL2/COL2-derived peptides using animal models [41,178] (Table 1). 

The ultimate aim of these experiments is to establish antigen-specific tolerance that ameliorates and prevents disease development. The major role of glycosylation of K264 in the COL2 T-cell immunodominant epitope has been highlighted by experiments showing effective therapeutic vaccination that inhibited disease development in mice with CIA. This approach was based on intravenous administration of soluble complexes of MHC II molecules with bound glycosylated COL2_259–273_ [41]. Ten years after this study, an improved method for COL2-specific tolerance induction in a humanized DR4-expressing mouse model was reported based on application of nanoparticles conjugated with MHC II molecules in a complex with COL2_259–273_ peptide [44]. However, the collagen peptides used in these experiments were not glycosylated and the reported effects were weaker compared to the results with glycosylated peptides, thus strengthening the importance of K264 posttranslational modification. This hypothesis is supported by more recent investigations that displayed the long-term amelioration of arthritis symptoms in mice following the therapeutic intravenous administration of fructosylated at K264 peptides derived from bovine COL2 [181]. Although, the sugar residue attached to K264 is slightly different in these experiments, the obtained results strongly support the general role of glycosylation as an important PTM of the T-cell immunodominant epitope from COL2 with potential for the development of new therapeutic approaches for RA.

Another approach based on COL2 PTMs is the development of a multiepitope product (ME-Cit) containing citrullinated peptides from vimentin, fibrinogen, filaggrin and COL2 [182]. The therapeutic application of ME-Cit ameliorated disease symptoms in rats with adjuvant-induced arthritis and induced higher levels of regulatory T-cells, while the magnitude of Th17 responses was decreased [182]. Subsequently, the results were confirmed using peripheral blood mononuclear cells (PBMCs) from early diagnosed RA patients [183]. It was shown that citrullinated peptides can modulate the expression of pro- and anti-inflammatory cytokines from RA PBMCs, regulatory T-cells and Th17 populations, providing evidence that citrullinated multiepitope peptides might be used as immunomodulatory agents in antigen-specific therapeutic approaches.

Several diagnostic strategies based on citrullinated COL2 peptides were reported (Table 2). 

In 2012, Hansson et al. reported the establishment and validation of a multiplex chip-based assay for the detection of antibodies specific to different citrullinated proteins and their unmodified arginine-containing variants [184]. The assay allowed for the analysis of more than 100 different antibody specificities. The peptides used represented epitopes from five candidate autoantigens—fibrinogen, α-enolase, vimentin, filaggrin and COL2—as well as multiepitope peptides that included the citrullinated C1 epitope of the COL2 sequence. The assay was tested on a cohort of 927 newly diagnosed RA patients (within 12 months of manifestation of first disease symptoms) and 461 controls and shown to have prognostic and diagnostic value, provide sample-saving testing and potential guidance for the choice of a personalized treatment strategy [184]. Another study concentrated on cyclic citrullinated COL2 peptides, validating their application for diagnostic purposes because RA patient samples showed higher reactivity compared to osteoarthritis patients and healthy controls [102]. In addition, novel cit-COL2 epitopes were identified, and antibodies against them showed a heterogeneous binding pattern. 

Antibodies recognizing citrullinated COL1 and COL2 telopeptides have been shown to have predictive value for the development of seropositive RA when combined with the detection of antibodies to citrullinated mutated vimentin, and were detected years before disease onset [185]. All these data cumulatively confirm the diagnostic value of citrullinated COL2 peptides. Importantly, to significantly improve diagnostic assays, a combination with other citrullinated, as well as carbamylated and acetylated epitopes, is needed. Assays based on the detection of antibodies against different posttranslationally modified COL2 epitopes could provide higher sensitivity and facilitate the stratification of different RA patients’ subgroups.

## 4. Conclusions

Immunity against posttranslationally modified proteins plays a significant role in RA pathogenesis. The definition of risk factors and the specific locations of some PTMs reactions in the organism contributed to the development of an improved hypothesis for the mechanism of RA induction. It is now generally accepted that the initial arthritogenic reactivities directed towards posttranslationally modified proteins originate from mucosal sites. Subsequently, the AMPA responses could develop at/attack different sites during different time points in RA development. The affinity maturation of AMPA-producing B-cells achieved by somatic hypermutations contributes to epitope spreading and the polyreactivity of pathogenic antibodies, eventually targeting joints and leading to destruction and disability. In addition, AMPA responses can display broad reactivity expanding beyond one single PTM type and including other PTMs. Furthermore, joint inflammation primes chemical reactions and provides altered conditions that modify the major protein in hyaline cartilage—COL2. COL2 PTMs may contribute to RA chronicity as they provide new epitopes during the inflammation process, which continuously drive autoimmunity and joint destruction. Therefore, the fine determination of anti-modified COL2 reactivity could contribute to RA subgroups’ stratification, improved diagnosis and personalized therapy.

The opportunity to diagnose RA at a preclinical stage would prevent patient suffering and reduce social and economic burden caused by the disease. The PTMs of various tissue-specific and ubiquitous proteins have contributed towards a notable degree of development in this regard. However, recent experimental evidence suggests a complex role of antibody responses to restricted COL2 PTMs in preclinical RA, which requires further investigation. The detection of different combinations of anti-PTM responses directed against different antigens would improve diagnostic assays’ sensitivity and specificity and allow the prediction of the course of the disease. Multiplex immunoassays that include the detection of citrullinated and carbamylated COL2 epitopes in combination with an analysis of reactivity against other posttranslationally modified autoantigens seem to be the future of new diagnostic strategies. Putative examples for other PTM autoantigens are fibrinogen, vimentin, filaggrin, etc., but also tenascin-C. Autoimmune responses directed towards citrullinated tenascin-C peptides have been recently associated with RA pathogenesis [23,186]. Therefore, the multiplex detection of autoantibodies against COL2 PTM epitopes and other specific epitopes, including citrullinated tenascin-C peptides, could significantly improve RA diagnosis.

The complex nature of RA is strong evidence for the need for a personalized immune-specific therapeutic approach for patients that do not respond to the generally applied therapy. Such strategies would eliminate the adverse effects of the present biologic agents, such as the general suppression of immune responses. The research performed to date suggests that patients with detected reactivity against COL2 PTMs would benefit from antigen-specific tolerogenic therapy that applies modified COL2 epitopes. 

## Figures and Tables

**Figure 1 ijms-24-09884-f001:**
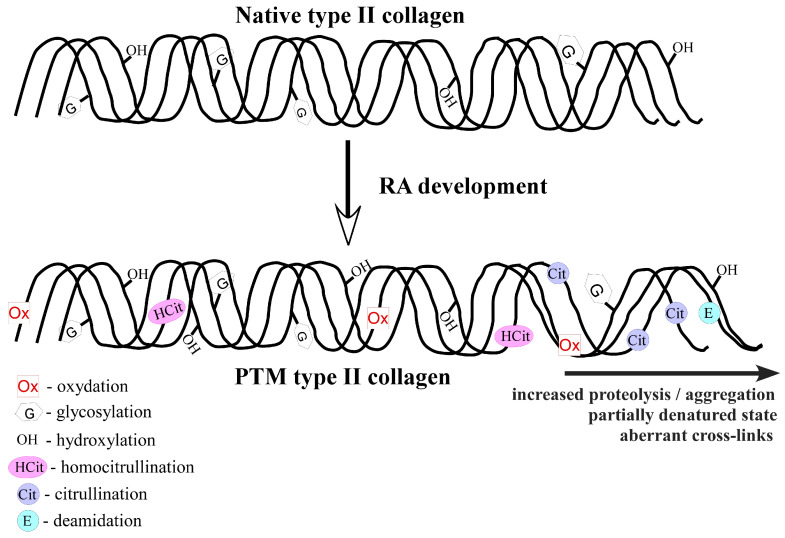
Types of collagen type II posttranslational modifications associated with rheumatoid arthritis.

**Figure 2 ijms-24-09884-f002:**
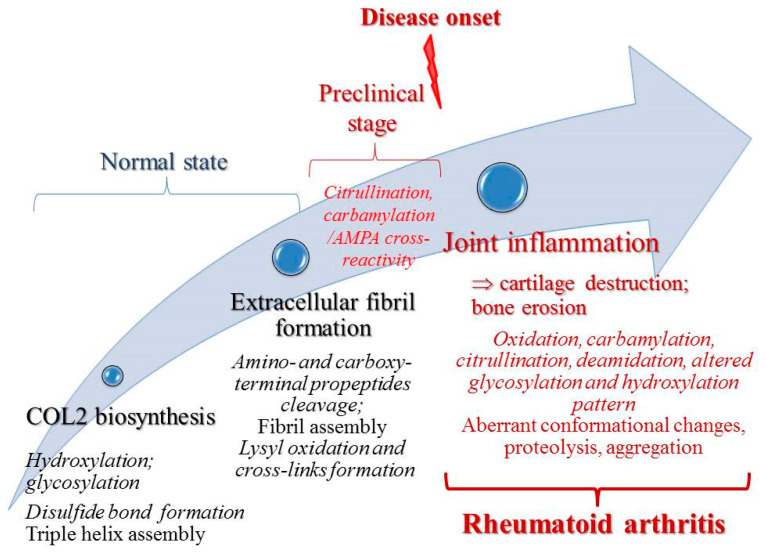
Overview of COL2 PTMs from biosynthesis and normal physiological state to rheumatoid arthritis.

**Table 1 ijms-24-09884-t001:** Therapeutic strategies based on COL2 PTMs.

PTM Type	Test Objects	Effect
Glycosylated COL2_259–273_	B10.Q mice/ (BALB/c × B10.Q) F2 CIA model	Inhibition of CIA incidence and severity; reduced anti-CII antibody levels Reduction in arthritis progression in chronic stage
Fructosylated COL2_259–273_	DBA/1 mice FIA-CIA model *	Amelioration of disease severity No effect on antibody levels
Citrullinated multiepitope peptide containing COL2 sequence	Lewis rats AIA model *	Ameliorated disease Increased regulatory T-cell populations Reduced Th17 populations
	PBMC from RA patients	Increased regulatory T-cell populations and TGF-β expression levels Reduced Th17 populations; downregulation of IL-1β and TNF-α expression

* FIA-CIA: fibrinogen-induced arthritis–collagen-induced arthritis; AIA: adjuvant-induced arthritis.

**Table 2 ijms-24-09884-t002:** Diagnostic assay approaches involving COL2 PTMs.

Detected Antibody Specificity	Results/Benefits
Citrullinated carboxy-terminal COL1 and COL2 telopeptides (TELO-I and TELO-II, respectively); Mutated citrullinated vimentin (MCV)	Prediction of seropositive RA
Citrullinated C1 epitopes of COL2 and multiepitope peptides; 11 other citrullinated peptides containing epitopes from α-enolase, vimentin, fibrinogen, filaggrin	Improved diagnostic value Potential guidance for personalized treatment
Cyclic citrullinated COL2 peptides	Identification of new epitopes Improved specificity for RA diagnosis

## Data Availability

Not applicable.

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
