# Peer review of "Significance of Type II Collagen Posttranslational Modifications: From Autoantigenesis to Improved Diagnosis and Treatment of Rheumatoid Arthritis"

_ijms, 2023, doi:10.3390/ijms24129884_

Round 1

Reviewer 1 Report

In this well-written work Authors focused on the COL2 posttranslational modifications as markers for the diagnosis of Rheumatoid Arthritis, for the monitoring of pre-clinic stages of the disease, and for the design of specific and personalized therapeutic strategies.

When in the Conclusions section authors state "Multiplex immunoassays that include......seem to be the future of new diagnostic strategies" (560-562), do they refer to tenascin- C and/or to other types of Collagens? Please,  insert a comment about the PTMs of Tenascin-C  that might improve the diagnostic assays together with COL2 PTMs' panel.

I suggest the insertion of a  graphycal abstract.

Author Response

We thank the reviewer for thorough evaluation of our work. We appreciate the constructive comments and valuable suggestions of the reviewer and the manuscript has been revised accordingly. Shown below are our point-by-point responses.

When in the Conclusions section authors state "Multiplex immunoassays that include......seem to be the future of new diagnostic strategies" (560-562), do they refer to tenascin- C and/or to other types of Collagens? Please, insert a comment about the PTMs of Tenascin-C that might improve the diagnostic assays together with COL2 PTMs' panel.

In this sentence we refer to PTM epitopes from different proteins including tenascin-C. The Conclusions section has been supplemented.

I suggest the insertion of a graphycal abstract.

A graphical abstract has been included.

Reviewer 2 Report

This is a very interesting review on the role of type II Collagen post-translational modification in the pathogenesis, the diagnosis and treatment of rheumatoid arthritis. 

The review is well written and organised and it is based on the most recent literature data

I have no further revision to suggest 

Author Response

We are grateful to the reviewer for the positive comments!

Round 2

Reviewer 1 Report

Now the manuscript is accepted